# Hydride-based antiperovskites with soft anionic sublattices as fast alkali ionic conductors

Shenghan Gao[1,6], Thibault Broux[1,6], Susumu Fujii [2,6], Cédric Tassel [1✉], Kentaro Yamamoto[3], Yao Xiao[3], Itaru Oikawa[4], Hitoshi Takamura [4], Hiroki Ubukata[1], Yuki Watanabe[1], Kotaro Fujii [5], Masatomo Yashima [5], Akihide Kuwabara [2], Yoshiharu Uchimoto [3] & Hiroshi Kageyama [1✉]

Most solid-state materials are composed of p-block anions, only in recent years the introduction of hydride anions ($1s^2$) in oxides (e.g., $SrVO_2H$, $BaTi(O,H)_3$) has allowed the discovery of various interesting properties. Here we exploit the large polarizability of hydride anions ($H^-$) together with chalcogenide ($Ch^{2-}$) anions to construct a family of antiperovskites with soft anionic sublattices. The $M_3HCh$ antiperovskites (M = Li, Na) adopt the ideal cubic structure except orthorhombic $Na_3HS$, despite the large variation in sizes of M and Ch. This unconventional robustness of cubic phase mainly originates from the large size-flexibility of the $H^-$ anion. Theoretical and experimental studies reveal low migration barriers for $Li^+$/$Na^+$ transport and high ionic conductivity, possibly promoted by a soft phonon mode associated with the rotational motion of $HM_6$ octahedra in their cubic forms. Aliovalent substitution to create vacancies has further enhanced ionic conductivities of this series of antiperovskites, resulting in $Na_{2.9}H(Se_{0.9}I_{0.1})$ achieving a high conductivity of ~$1 \times 10^{-4}$ S/cm (100 °C).

[1] Department of Energy and Hydrocarbon Chemistry, Graduate School of Engineering, Kyoto University, Nishikyo-ku, Kyoto 615-8510, Japan. [2] Nanostructures Research Laboratory, Japan Fine Ceramics Center, Nagoya 456-8587, Japan. [3] Graduate School of Human and Environmental Studies, Kyoto University, Sakyo-ku, Kyoto 606-8501, Japan. [4] Department of Materials Science, Graduate School of Engineering, Tohoku University, Sendai 980-8579, Japan. [5] Department of Chemistry, School of Science, Tokyo Institute of Technology, 2-12-1-W4-17 O-okayama, Meguro-ku, Tokyo 152-8551, Japan. [6] These authors contributed equally: Shenghan Gao, Thibault Broux, Susumu Fujii. ✉email: cedric@scl.kyoto-u.ac.jp; kage@scl.kyoto-u.ac.jp

Perovskites with a formula of $ABX_3$ (A, B: cations, X: anions) are ubiquitous and central to electronics, photonics, and energy technologies[1,2]. Their electronically inverted analogs, antiperovskites, are playing an increasingly major role in solid-state chemistry and physics owing to their various intriguing properties, such as giant magnetoresistance in $Mn_3GaC$[3], superconductivity in $Ni_3MgC$[4], negative thermal expansion in $Mn_3BA$ (B = Zn, Cu etc.; A = N, C)[5,6], efficient water splitting electrocatalytic activity in $Ni_3FeN$[7], superionic conductivity in $Ag_3SI$[8] and large capacity in $(Li_2Fe)OCh$ (Ch = S, Se, Te) as lithium battery cathodes[9].

In particular, lithium-rich and sodium-rich antiperovskites (LiRAPs and NaRAPs), for example, $M_3OCl$, $M_3OBr$ (M = Li, Na), and $Na_3OBH_4$, have recently attracted a great deal of attention as they exhibit a high lithium (or sodium) ionic conductivity and are thus regarded as promising solid-state electrolytes enabling high-energy-density lithium metal batteries[10–13]. Similar to the $F^-$ superionic conductivity in $NaMgF_3$ perovskite[14], the antiperovskite $M_3OCl$ allows $Li^+/Na^+$ superionic conductivity benefited from the M-rich content (60 at% in $Li_3OCl$)[10,13]. Another advantage of antiperovskite is its extraordinary chemical diversity. Similar to perovskite, a variety of combinations of elements can be accommodated, while maintaining the simple structural topology, thereby offering an ideal situation to easily and fully characterize fast ionic transport[15,16]. A very recent study on $Li_3OCl$ indicated that the presence of hydroxide ($OH^-$), providing the composition of $Li_{3-x}O_{1-x}(OH)_xCl$, or ultimately $Li_2(OH)Cl$[17].

In the search of new fast ion-conducting solid-state materials[18–27], a great deal of effort has been made to understand ionic transport in Li-ion conductors, where two features of the anion-host matrix have been highlighted[18–20]. The first feature proposed by Ceder et al. is that the anion arrangement of body-centered cubic (bcc) provides the lowest migration barrier for Li-ion diffusion, rather than a close-packed (fcc or hcp) anion framework[27]. Antiperovskites host the bcc framework composed of A- and B-site anions[10,13,16]. The second feature is that polarizable anions can critically soften and flatten the cationic transport landscape, leading to lower activation energy and higher ionic conductivity[23,25,28–31]. In fact, thiophosphate ionic conductors, such as $Li_{10}GeP_2S_{12}$ with the highest Li-ion conductivity, are believed to benefit from a soft and polarizable anion lattice[32].

In this study, we present the synthesis of a series of LiRAPs and NaRAPs, $M_3HCh$ (M = Li, Na; Ch = S, Se, Te) with both anionic sites occupied by soft and polarizable anions of $H^-$ and $Ch^{2-}$ (Fig. 1)[33,34]. To our knowledge, LiRAPs and NaRAPs have been obtained only with oxide/hydroxide anions at the B site, which considerably limits the scope of structures and properties. Experimental and theoretical investigations have revealed that the occupation of highly polarizable and size-flexible $H^-$ anions at the B site in $M_3HCh$ introduces several interesting features, including the stability and robustness of the ideal cubic structure and softening of phonon mode associated with $HM_6$ octahedral rotation, which could be advantageous for realizing high $Li^+/Na^+$ conductivity. The energy migration barriers based on vacancy and interstitial dumbbell mechanisms are found to be lower than those of oxide-based LiRAPs and NaRAPs. $Li^+/Na^+$ vacancy creation via iodine ($I^-$) doping (for $Ch^{2-}$) is a potential strategy to increase the ionic conductivity of this series of antiperovskites, where I-doped cubic $Na_3HSe$ with formula $Na_{2.9}H(Se_{0.9}I_{0.1})$ delivers a high total $Na^+$ conductivity exceeding $1 \times 10^{-4}$ S/cm at 100 °C with a low bulk activation energy of ~0.18 eV in agreement with the calculated one (~0.16 eV).

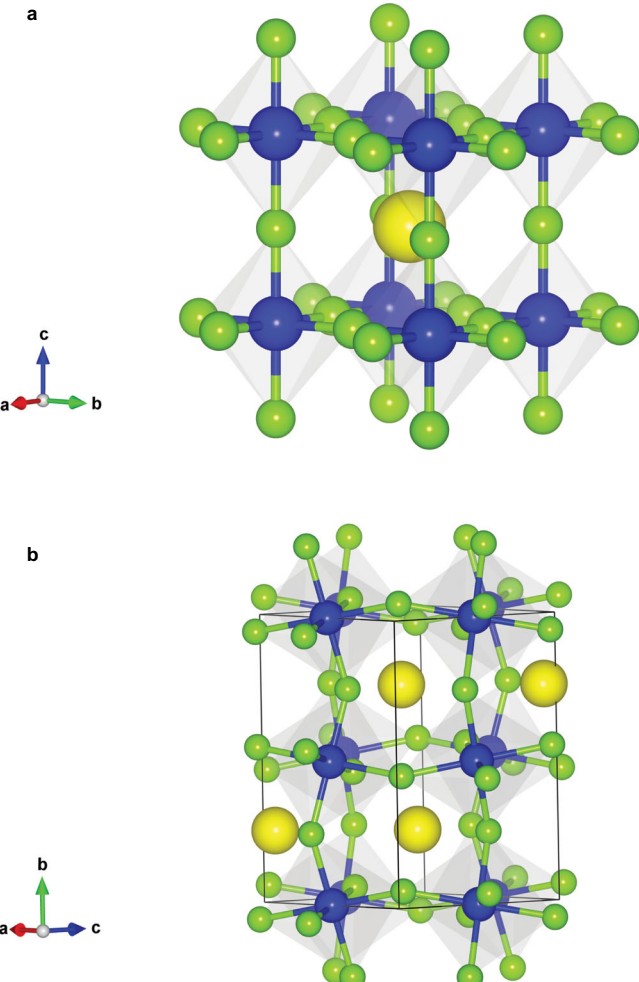

**Fig. 1 Crystal structure of hydride-based M3HCh antiperovskite (M = Li, Na). a** Crystal structure of the cubic antiperovskite (space group $Pm$–$3m$). In this study, we obtained $Li_3HCh$ (Ch = S, Se, Te) and $Na_3HCh$ (Ch = Se, Te), where $H^-$ anion (blue; B site) is bonded with $Li^+/Na^+$ cations (green) forming $HLi_6/HNa_6$ octahedron, while chalcogenide anion (yellow; A site) sits in cuboctahedral coordination site. As opposed to conventional perovskites, the cubic structure is robust in a wide range of compositions. **b** The orthorhombic anti-$GdFeO_3$ type (space group $Pnma$) $Na_3HS$ with $HNa_6$ octahedral tilting.

## Results

**Structure determination.** The high-resolution synchrotron X-ray diffraction (SXRD) pattern of the sample prepared using LiH and $Li_2S$ at 5 GPa and 700 °C (Fig. 2a) shows the formation of a highly crystalline compound, along with impurity phases that could be identified as LiH, $Li_2S$, and BN (insulating high-pressure sleeve). The diffraction profile of the main phase was indexed in a cubic system with the lattice parameter of $a = 3.85189(6)$ Å (see Supplementary Fig. 1), which is comparable to those of reported antiperovskite oxyhalides $Li_3OCl$ and $Li_3OBr$ ($a = 3.900$ Å and 3.989 Å, respectively) with the space group $Pm$–$3m$[10,12].

Given the known antiperovskites, e.g., $Ag_3SI$[8], $Li_3OCl$[10], with a smaller anion at the octahedral B site and a larger one at the cuboctahedral A site, we performed Rietveld refinement assuming $S^{2-}$ at the $1b$ (1/2, 1/2, 1/2) Wyckoff position and $H^-$ at the $1a$ (0, 0, 0) site and $Li^+$ at the $3d$ (1/2, 0, 0) site within the space group of $Pm$-$3m$, corresponding to a stoichiometric $Li_3HS$ formula. The Rietveld refinement converged successfully with values of

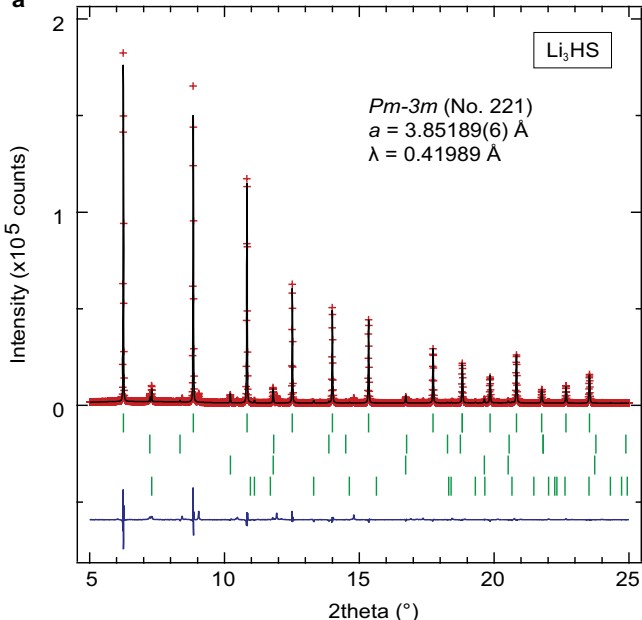

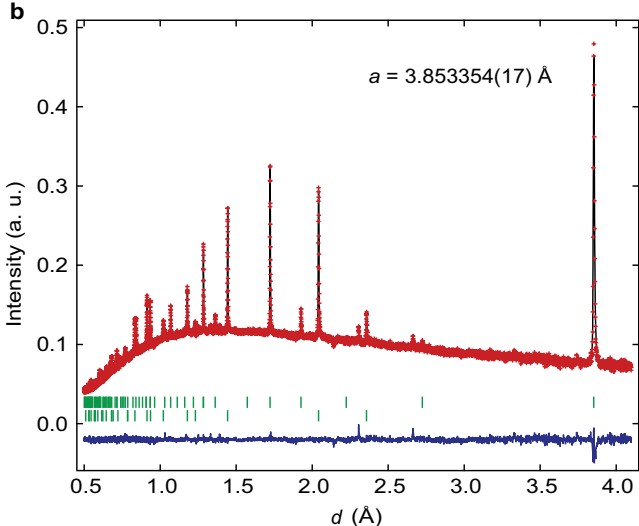

**Fig. 2 Structural determination of Li₃HS.** Rietveld refinement of **a** SXRD and **b** ND. The red crosses, black solid line, blue solid line, and green dashes denote, respectively, the observed, calculated, difference intensities, and calculated Bragg reflections (from top to bottom: Li₃HS, Li₂S, LiH, BN in **a**; Li₃HS, LiH in **b**).

$R_{\mathrm{Bragg}} = 4.28\%$ and $R_f = 2.87\%$ (see Fig. 2a and Supplementary Table 1). Swapping the anionic octahedral B site and cubocathedral A site to give a Li₃SH led to poor refinement values of $R_{\mathrm{Bragg}} = 22.6\%$ and $R_f = 11.2\%$ (Supplementary Fig. 2).

Subsequently, time-of-flight powder neutron diffraction (ND) data were analyzed to better characterize lighter elements of H and Li. The presence of hydrogen in the sample can be readily seen from Fig. 2b exhibiting a high background intensity. The neutron refinement assuming the Li₃HS structure yielded $R_{\mathrm{wp}} = 1.35\%$ and $R_{\mathrm{Bragg}} = 6.28\%$ thus confirming this structural model. Details of the refinement are listed in Supplementary Table 1. It is found that the Li⁺, H⁻, and S²⁻ ions fully occupy their respective crystallographic sites, confirming the stoichiometric composition. We also examined the possibility of antisite disorder between H⁻ and S²⁻; the Rietveld refinement allowing intersite anion exchange

(Supplementary Table 1) in the stoichiometric composition led to the full occupancy ($g$) of H⁻ at the B site ($g_B(H) = 1.015(5)$) with $R_{\mathrm{wp}} = 1.37\%$ and $R_{\mathrm{Bragg}} = 6.64\%$, which implies that the anions are perfectly ordered. Note that Pd₃H₀.₈₉In also has hydrogen at the octahedral site[35], but this material may not be classified as hydride, given the metallic nature of this material and the nearly identical electronegativity between H (2.2) and Pd (2.2)[33]. Attempts to synthesize Li₃HS compound at a lower pressure (1 GPa/3 GPa; 700 °C) or under ambient pressure (700 °C for 12 h) using LiH and Li₂S was unsuccessful (Supplementary Fig. 3), indicating the high-pressure metastability of this antiperovskite.

**Materials variety.** To our knowledge, Li₃HS is the first LiRAP with hydride anions at the B site. Next, we attempted to extend the antiperovskite family by including sodium as well as other chalcogenides under high-pressure and high-temperature reactions. The XRD patterns of Li₃HSe and Li₃HTe (Supplementary Fig. 4) are similar to Li₃HS, with Bragg reflections moving toward lower angles, as expected from increasing ionic radii of chalcogenide anions. The obtained cell parameters of the cubic unit cell are 3.9744(5) Å for Li₃HSe and 4.2221(3) Å for Li₃HTe. Regarding the sodium system, the SXRD profiles of Na₃HSe (Supplementary Fig. 5) and Na₃HTe (Supplementary Fig. 6) are compatible with the cubic symmetry (*Pm–3m*) with $a = 4.55901(7)$ Å and $a = 4.76349(2)$ Å, respectively. The larger cell parameters result from the difference in the cationic size (Na⁺: 1.02 Å vs. Li⁺: 0.76 Å)[36]. Rietveld refinements of Na₃HSe and Na₃HTe validated the cubic antiperovskite structure, with detailed structural information in Supplementary Tables 2 and 3.

To gain microscopic information on the crystal structure, we conducted ²³Na nuclear magnetic resonance (NMR) with/without magic-angle spinning (MAS) for Na₃HSe. The spectra in Supplementary Fig. 7 were fitted as a second-order quadrupolar line shape of the central transition with a common set of $C_Q = 1.61$ MHz and $\eta = 0.04$, where $C_Q$, and $\eta$ denote a quadrupole coupling constant and an asymmetry parameter, respectively. The ²³Na MAS NMR spectrum shows a single sharp peak at 24.2 ppm, which indicates that all sodium atoms are in the same environment, in accordance with the refinement result that no significant chemical disorder occurs in Na₃HSe (Supplementary Fig. 5 and Supplementary Table 2). This isotropic chemical shift (i.e., 24.2 ppm) is fairly consistent with that derived from DFT calculation (20.4 ppm with $C_Q = 1.95$ MHz and $\eta^{\mathrm{DFT}} = 0$).

In the case of Na₃HS, the XRD pattern (Supplementary Fig. 4) is clearly different and indexed by an orthorhombic unit cell ($a = 6.76037(9)$ Å, $b = 8.89761(10)$ Å, and $c = 6.28659(8)$ Å), which is related to the pristine cubic cell by $\sqrt{2}a_p \times 2 b_p \times \sqrt{2}c_p$. This supercell and the extinct reflections suggest that Na₃HS adopts an anti-GdFeO₃ structure (*Pnma* space group, Fig. 1b), as previously reported for M₃OA (M = Sr, Eu, Ba, and A = Si, Ge)[37]. SXRD (Supplementary Fig. 8) and ND (Supplementary Fig. 9) data were refined assuming the anti-GdFeO₃ structure, yielding reasonable parameters, as listed in Supplementary Tables 4 and 5.

As shown in Fig. 3a, the normalized (cubic) lattice parameters of the LiRAP and NaRAP series both show linear dependence as a function of the chalcogen ionic radii, with approximately the same slope. As it is commonly done for normal perovskite structures, the tolerance factor ($t$) was estimated for our system, assuming the hydride ionic radius of 1.40 Å[36,38]. We found that the cubic structures are observed over a wide range $0.85 < t < 0.97$ (Supplementary Table 6), in sharp contrast to perovskite oxides where slight deviations from unity readily lead to structural distortions. It is also noticed that oxide-based antiperovskites M₃OA (M = Ca, Sr, Ba, Eu; A = Si, Ge, Sn, Pb) exhibit octahedral tilting when $t < 0.97$[37], implying that our hydride-based system is

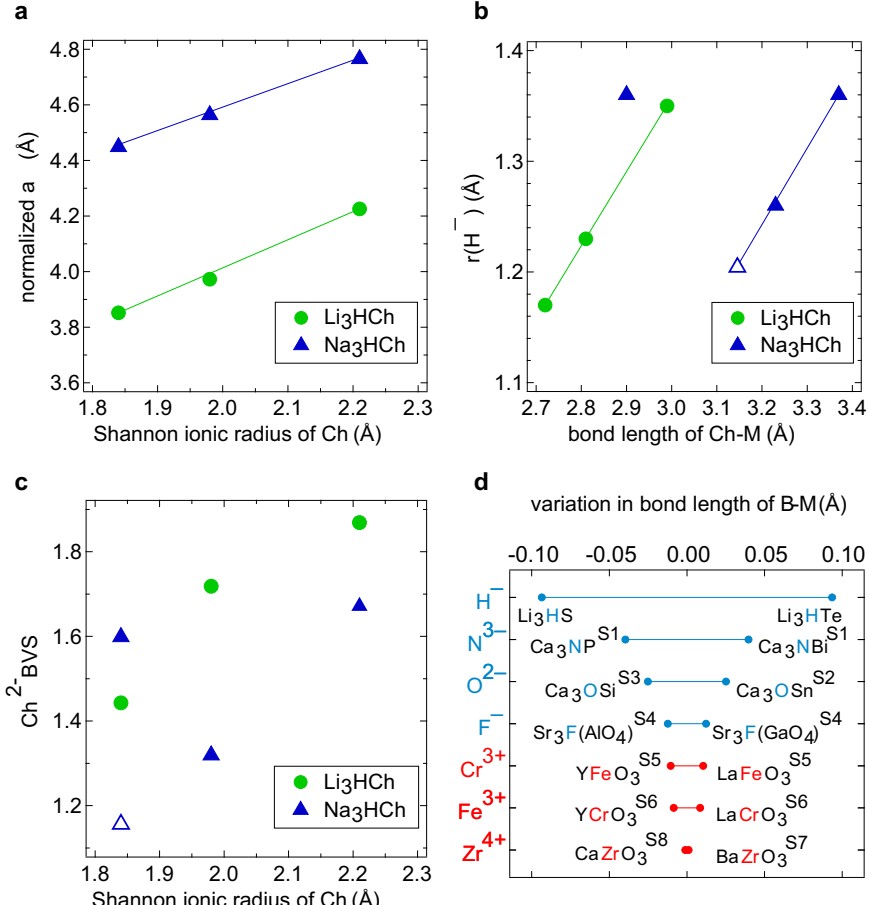

**Fig. 3 Structural features of M₃HCh antiperovskites (M = Li, Na). a** Normalized (cubic) lattice parameters of M₃HCh. For orthorhombic o-Na₃HS, the cubic lattice parameter is an average of normalized lattice parameters. Lines are for the eye guidance. **b** The hydride anion (H⁻) size obtained by subtracting M⁺ radius from H⁻–M⁺ bond length. Open triangles indicate the ionic radius of H⁻ in hypothetical cubic c-Na₃HS, which is calculated by using the average normalized cubic lattice parameter of the orthorhombic o-Na₃HS. **c** BVS values for Ch²⁻. Open triangles indicate the S²⁻ BVS for hypothetical cubic c-Na₃HS. **d** Variation in bond length of B-M in BM₆ (B = H⁻, N³⁻, O²⁻, F⁻) octahedra for M₃BA antiperovskites and B-O in BO₆ (B = Fe³⁺, Cr³⁺, Zr⁴⁺) octahedra for ABO₃ perovskites. The mentioned studies refer to Supplementary References (S1–S8).

quite anomalous. The *t* value of the orthorhombic Na₃HS is 0.84, slightly smaller than that of cubic Li₃HS (*t* = 0.85).

**Phase stability**. Earlier studies on oxyhydrides have shown that the hydride anion (H⁻) adapts itself to the different local environments, resulting in the hydride size flexibility; this unique feature brings about a number of novel properties such as high-pressure-induced coordination reversal in LaHO[33,39–42]. This means that our assumption of the fixed hydride size (1.40 Å) for calculating the tolerance factor was not adequate. Hence, we estimated the ionic radius of hydride anions, r(H⁻) by subtracting the ionic radii of alkali metals, r(M⁺) from the H–M bond lengths (a/2 in the case of the cubic system). Figure 3b shows that the hydride substantially changes its size, with r(H⁻) ranging from 1.17 Å to 1.36 Å. For the Li₃HCh series, r(H⁻) expands nearly linearly with the Ch–M bond length down the chalcogen group, in line with the increasing volume of HLi₆ octahedron and ChLi₁₂ cuboctahedron (Supplementary Fig. 10). A similar linear dependence can be recognized as long as the cubic phase of Na₃HCh (including the hypothetical cubic c-Na₃HS) is considered. Interestingly, the hydride size in the real orthorhombic o-Na₃HS deviates from this linear relationship; r(H⁻) in o-Na₃HS greatly increases to the value closer to cubic Na₃HTe. The particularly

soft anion (H⁻) with the flexible size is counterintuitive to the traditional hard-sphere model with fixed ionic radii when describing the bonding nature of ionic compounds.

To better understand the swollen H⁻ in orthorhombic o-Na₃HS, we calculated bond valence sum (BVS) values of chalcogenide using the tabulated parameters[43]. Note that the size flexibility of hydride anion does not permit a reliable estimate of BVS for the hydride anion itself[40]. As shown in Fig. 3c, the BVS value for S²⁻ in hypothetical c-Na₃HS is –1.16, which is unusually low, indicating that S²⁻ is greatly underbonded in the cuboctahedral site. Here, the flexible hydride comes into play. In o-Na₃HS, the HNa₆ octahedron (due to swollen H⁻) is greatly expanded (as displayed in Supplementary Fig. 10), which in turn reduces the SNa₁₂ volume and allows S²⁻ to gain an acceptable BVS value of –1.60. For the Li₃HCh series, Li₃HS has the lowest Ch-BVS value of –1.44, which is higher than that for Na₃HSe (–1.32).

Since we have estimated r(H⁻) for each compound (Fig. 3b), we can redefine the tolerance factor (*t'*) (see Supplementary Table 6). It is found that cubic hydride antiperovskites have large *t'* from 0.93 to 0.99, while orthorhombic Na₃HS has *t'* = 0.85. The redefined *t'* for Li₃HCh is closer to unity and has a narrower range. For example, Li₃HS with a minimum *t* of 0.85 changes to *t'* = 0.95 when the observed r(H⁻) is applied. Therein lies the

extraordinary size flexibility of hydride, that is, the marked variation in bond length of H–Li in $HLi_6$ octahedra, from a maximum of 2.11 Å for $Li_3HTe$ to a minimum of 1.93 Å for $Li_3HS$, as shown in Fig. 3d. The obtained bond-length difference of 0.09 Å for hydride antiperovskites is much larger than those of other antiperovskites with B = N (0.04 Å), O (0.03 Å), and F (0.01 Å). However, from a broader perspective, antiperovskites generally have larger octahedral size variations than conventional oxide perovskites, developing the potential to tailor structures toward acquiring new functions.

Until now, we had a sharp picture of the large size variation of hydride anion. The chalcogenide anion, however, is also polarizable. To critically evaluate the role of anions in stabilizing the cubic structure, we estimated the radius of each ion using Bader population analysis, which partitions the first-principles-calculated charge density grid into the Bader region of each ion (see Methods in Supplementary Information)[44]. The resulting Bader radius of $Li^+$ only increases from 0.94 to 0.97 Å when the A-site chalcogenide anion increase from $S^{2-}$ to $Te^{2-}$, whereas the Bader radius of $H^-$ ion expands from 1.38 to 1.48 Å (Supplementary Table 7). The large size variation of $H^-$ is also evidenced in $Na_3HCh$. Interestingly, the change in Bader radius of $Ch^{2-}$ ions is comparable to that of $H^-$ ions; the Bader radii of $Se^{2-}$ and $H^-$ are 2.06 and 1.42 Å, respectively, in $Li_3HSe$ and increase to 2.24 and 1.52 Å in $Na_3HSe$. However, when applying the external pressure (5 GPa) to $M_3HCh$, the Bader radius of $H^-$ decreases more substantially than that of $Ch^{2-}$ (Supplementary Fig. 11). The hydride ion is more sensitive to the applied pressure than chalcogenide. Given that most of $M_3HCh$ compounds are currently synthesized by high pressure, we argue that the flexible hydride is more critical for stabilizing the cubic symmetry.

To investigate the thermodynamic stability of this series of hydride antiperovskites, we evaluated the formation enthalpy $\Delta H$ of $M_3HCh$ (M = Li, Na) in the reaction $(MH + M_2Ch \rightarrow M_3HCh)$ under 0 and 5 GPa based on first-principles calculations, where $\Delta H$ is defined as $\Delta H = H(M_3HCh) - \{H(M_2Ch) + H(MH)\}$. Calculations showed that $Li_3HS$, $Na_3HS$, and $Na_3HSe$ are thermodynamically unstable at 0 GPa ($\Delta H > 0$), but are stabilized under high pressure (Fig. 4a). On the other hand, $Li_3HTe$ and $Na_3HTe$ with large $t'$ values (0.99 and 0.96, respectively) are stable even at ambient pressure. Hence, the proof-of-concept trial to synthesize $Na_3HTe$ was conducted by heating a mixed pellet of NaH and $Na_2Te$ at 400 °C overnight in a vacuum-sealed Pyrex tube. The resulting SXRD pattern (Supplementary Fig. 12) yielded a cubic phase with $a = 4.76717(5)$ Å, which is identical with the sample obtained under high pressure ($a = 4.76349(2)$ Å) and antisite anion disorder exists in neither case (Supplementary Table 8). In addition, the Gibbs free energy difference ($\Delta G$) showed that $Li_3HSe$ is stable at ambient pressure ($\Delta G = -0.01$ eV/f.u.), despite the slightly positive value of $\Delta H$ (Fig. 4a); our preliminary synthesis under simple ambient conditions has failed, but there remains room for adjusting parameters, such as partial $H_2$ gas pressure.

**Electron and phonon calculations**. The electronic band structures at 0 GPa (Supplementary Fig. 13) obtained using first-principles calculations show that all the compounds have relatively large bandgaps: 4.2 eV for $Li_3HS$, 3.5 eV for $Li_3HSe$, 2.9 eV for $Li_3HTe$, 2.9 eV for $Na_3HS$, 2.8 eV for $Na_3HSe$ and 2.7 eV for $Na_3HTe$. Although GGA-PBE exchange-correlation functional generally underestimates bandgaps[45], our compounds are expected to have electronically good insulating properties to meet the requirements as a solid electrolyte.

Phonon band structures of the hydride antiperovskites at 0 and 5 GPa were calculated using lattice dynamics within harmonic approximation (see Fig. 4b, c for $Li_3HS$ and Supplementary Fig. 14 for the others), including the hypothetical cubic $c$-$Na_3HS$. All the synthesized cubic compounds are found to be dynamically stable at 5 GPa. In contrast, the $c$-$Na_3HS$ under 0 and 5 GPa exhibits imaginary phonon modes at the M and R points, corresponding to in-phase and out-of-phase rotations of the $HM_6$ octahedral rotation, respectively[46], which as a result confirms the experimentally observed orthorhombic structure $o$-$Na_3HS$ with the $a^+b^-b^-$ tilting in Glazer notation. $Li_3HSe$, $Li_3HTe$, and $Na_3HTe$ are dynamically stable even at 0 GPa without any imaginary phonon frequencies. For $Li_3HS$ and $Na_3HSe$ (dynamically stable at 5 GPa), imaginary phonon modes (lattice instabilities) appear at M and R points under 0 GPa, which is inconsistent with experimental observations of the cubic phase after quenching from 5 GPa; this inconsistency might be resolved by including anharmonic effects of phonons at finite temperatures, which is the case of cubic $SrTiO_3$[47].

The calculated phonon dispersions of $M_3HCh$ (M = Li, Na) and atom-projected phonon DOS are quite unique; high frequencies associated with optical phonons are ascribed to local vibrations of the lighter $H^-$ ion, whereas vibrations by heavier $M^+$ and $Ch^{2-}$ ions are responsible for low phonon frequencies, along with relatively flat dispersions. Furthermore, the phonon band center of $M^+$ ions decreases as the $Ch^{2-}$ ion becomes larger (e.g., 9.6 to 9.2 to 8.4 THz for $Li_3HCh$), suggesting a tunability of lattice softness for mobile $M^+$ ions. Given the concept of "the softer the lattice, the better"[18,20,21,23,25,31] and the relatively large calculated bandgap, $M_3HCh$ antiperovskites should be a promising candidate as Li/Na solid-state electrolytes. In the next section, we will explore ionic transport properties in this series of hydride-based antiperovskites from a theoretical and experimental point of view.

**Ionic conductivity**. Nudged elastic band (NEB) calculations were used to estimate the energy barrier for cationic transport. Previous studies on $Li_3OCl$ proposed two transport mechanisms, i.e., Li hopping via vacancy mechanism and Li interstitial dumbbell hopping along edges of $LiO_6$ octahedra[10,12]. We calculated migration barriers for Li/Na transport in $M_3HCh$ based on these two mechanisms, and the results are displayed in Fig. 5a for the vacancy mechanism and Fig. 5b for the dumbbell mechanism. The energy barriers via vacancy mechanism range from 0.15 eV ($o$-$Na_3HS$, Supplementary Fig. 15) to 0.32 eV ($Li_3HTe$), while the dumbbell mechanism has a much smaller energy barrier from 0.05 eV ($Na_3HSe$) to 0.14 eV ($Li_3HTe$). The calculated energy barriers via both of mechanisms for $M_3HCh$ are notably low, and the difference of more than half between two mechanisms is also observed in previously reported antiperovskites (e.g., in $Li_3OCl$, 0.31 eV and 0.15 eV for vacancy and dumbbell mechanisms, respectively)[12]. It should also be noted that the calculated migration barrier (3.49 eV) for hydride ($H^-$) hopping via vacancy in $Li_3HS$ (Supplementary Fig. 16) is much higher than that of $Li^+$ (0.20 eV), indicating that $H^-$ anions hardly migrate in our antiperovskite. From the results of potentiostatic measurement of the symmetric cell $Li_3PS_4/Li_{2.9}H(S_{0.9}I_{0.1})/Li_3PS_4$, the steady-state current suggesting the sole motion of $Li^+$ in iodine-doped $Li_3HS$ is observed in Supplementary Fig. 17.

To our surprise, the energy barrier is largest for Ch = Te, followed by Se and S. This seems counterintuitive since the cell volume increases (corresponding to the widening of the channels for Li/Na transport) as the ionic radius of chalcogenide ion increase from S to Se and then Te. At the same time, the phonon band center of Li/Na becomes lower (enhancing the softness of lattice), which implies that other factors such as local lattice dynamics are playing a role in ion diffusion. A systematic

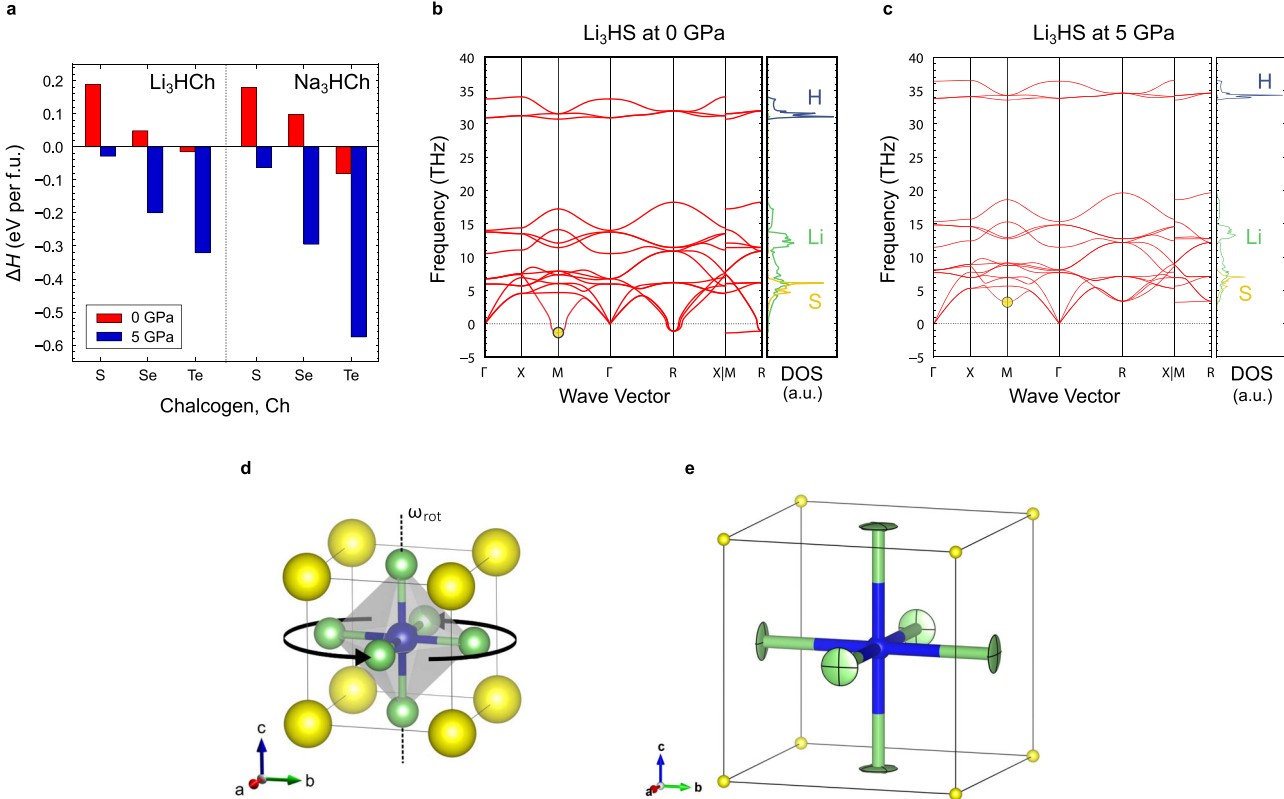

**Fig. 4 Formation enthalpy and phonon calculations. a** Formation enthalpy $\Delta H$ in the reaction $MH + M_2Ch \rightarrow M_3HCh$ (M = Li, Na) under 0 and 5 GPa. **b** Phonon band structures and partial phonon density of states (DOS) of Li, H, and S atoms in $Li_3HS$ under 0 GPa and **c** 5 GPa. The negative number of the vertical axis represents imaginary frequency. **d** $HM_6$ octahedral rotation associated with the phonon mode denoted by yellow circles in phonon band dispersion, where $\omega_{rot}$ is the frequency of the $HM_6$ rotational mode at M point. **e** Displacement ellipsoids of Li atoms depicted at 50% probability level obtained from Rietveld refinement of ND in $Li_3HS$.

investigation of phonon dispersions in $M_3HCh$ revealed that the rotational motion of the $HM_6$ octahedron reflecting on the phonon mode at M point (Fig. 4d) is essential to understand the unusual Ch dependence. This rotational motion corresponds well to the migration direction of $M^+$ ions to adjacent sites and displacement ellipsoids (anisotropic atomic displacement parameters) of Li/Na (Fig. 4e) obtained from the Rietveld refinement conform well to the rotational mode. The frequency of this phonon mode, $\omega_{rot}$, decreases with decreasing the $Ch^{2-}$ size (1.4$i$, 3.7 and 5.7 THz for $Li_3HS$, $Li_3HSe$ and $Li_3HTe$ and 0.6$i$ and 2.1 THz for $Na_3HSe$ and $Na_3HTe$), demonstrating the softening of this specific rotational phonon. For both Li/Na migration mechanism, $\omega_{rot}^2 \cdot m_M$ (where $m_M$ is the atomic mass of M), which corresponds to the force constant $k_{rot}$ with respect to the rotational motion, exhibits a positive linear correlation with energy barriers (Fig. 5c). The decreasing trend of $\omega_{rot}$ for Te $\rightarrow$ Se $\rightarrow$ S is directly related to the dynamic stability or the phase transition between the cubic phase and the tilted orthorhombic phase (Fig. 4 and Supplementary Fig. 14). Thus, the softening of $M^+$ ion migration (or octahedral rotational) mode could be the origin of the low energy migration energy of hydride antiperovskites.

Experimentally, we measured lithium/sodium-ion conductivity of cold-pressed $M_3HCh$ samples (M = Li, Na; Ch = S, Se, Te) using electrochemical impedance spectroscopy (EIS). As a representative example, we show in Supplementary Fig. 18 Nyquist plots of $Na_3HSe$, featuring typical ionic impedance response with a semicircle which has a capacitance of ~$10^{-10}$ F at high frequencies and a low-frequency tail. Since the bulk and grain boundary resistance cannot be specifically deconvoluted[48],

the total conductivity in Fig. 5d is attributed to the bulk and grain boundary resistance, which might be the reason why no clear trend in the composition dependence of ionic conductivity is observed. The activation energies obtained from the Arrhenius fit for the pristine $M_3HCh$ (Fig. 5d) are in the range of 0.44 to 0.53 eV for cubic phases and 0.30 eV for orthorhombic $Na_3HS$, which are comparable with other superionic conductors such as garnet $Li_7La_3Nb_2O_{12}$[49] and $\beta$-$Li_3PS_4$[50]. Compared with the computed migration barrier (Fig. 5a, b), the experimentally obtained activation energy (Fig. 5d) is relatively large. The discrepancy might result from two facts: the low concentration of intrinsic charge carriers (i.e., vacancies/interstitials) which we expect in the synthesized sample, whereas the calculated migration barriers assume intrinsic vacancies or interstitials of lithium/sodium. In addition, the migration barrier is separate from defect formation energy (which we list in Supplementary Table 9), whereas they are reflected in the experimental activation energies. Moreover, the computed compositional dependence (Fig. 5c) is not observed, probably due to the non-negligible resistive grain boundary and interfacial impacts on the conductivity measurement[51].

Here, we take the cubic $Na_3HSe$ compound as an example to fulfill the potential of the deformable anion matrix for fast sodium ionic diffusion. When creating a small number of sodium vacancies in $Na_{2.9}H(Se_{0.9}I_{0.1})$, the total ionic conductivity increased to ~$1 \times 10^{-4}$ S/cm at 100 °C with two orders of magnitude higher than the undoped one (Fig. 6a). It is also noteworthy that the Nyquist plots of I-doped $Na_{2.9}H(Se_{0.9}I_{0.1})$ sample at a lower temperature (Supplementary Fig. 19) show two semicircles at the high frequency where the small half-semicircle

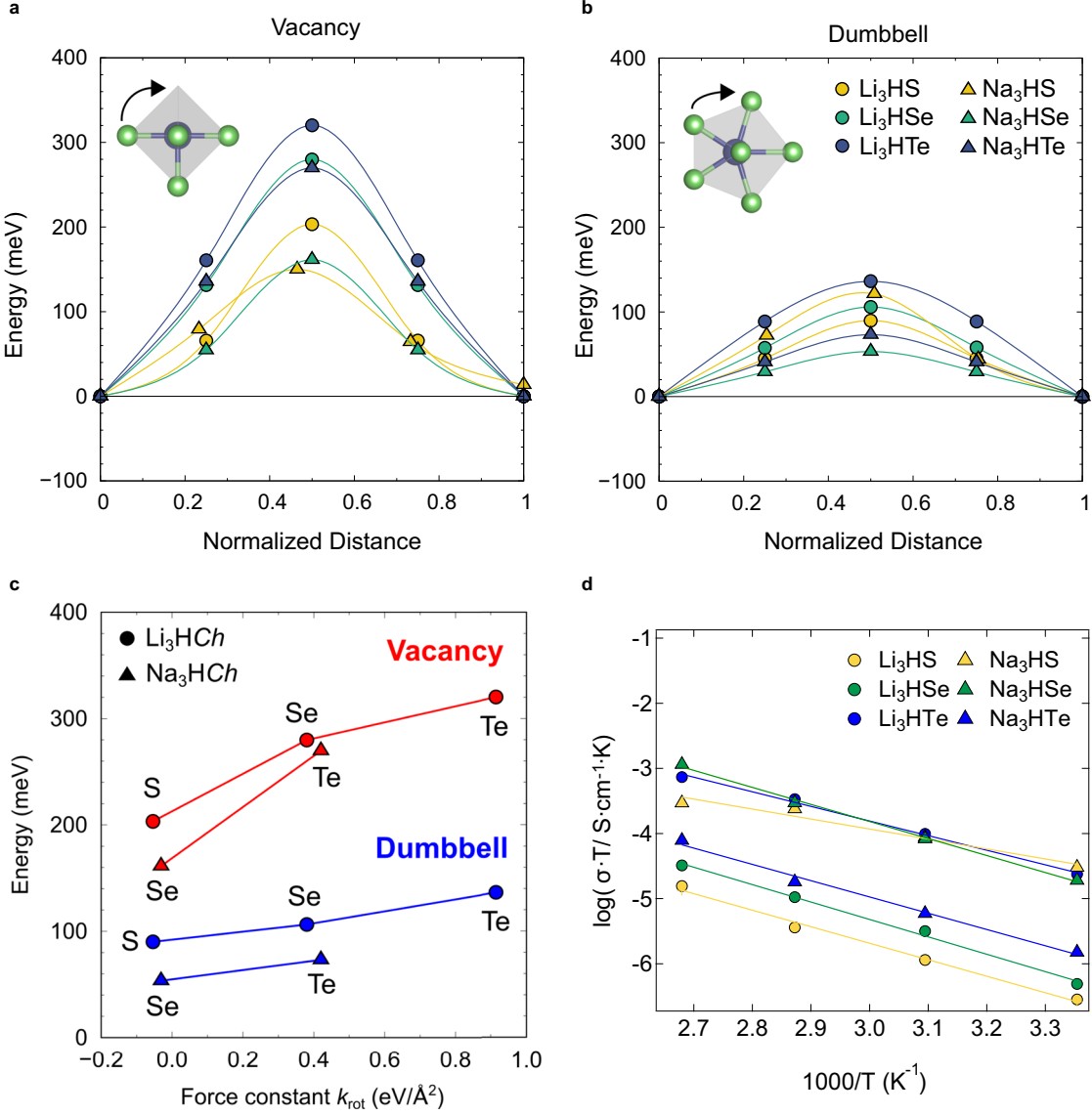

**Fig. 5 Ionic conductivities of M₃HCh antiperovskites (M = Li, Na, and Ch = S, Se, Te).** Low-barrier migration pathways for **a** vacancy and **b** interstitial dumbbell migration in $M_3HCh$. Top-left insets show $M^+$ migration when viewing $HM_6$ octahedron from the top to down. For orthorhombic $Na_3HS$, there are many migration pathways, one of which is shown as a representative example (see Supplementary Fig. 15 for details). **c** Migration barriers for cation transport via the vacancy and interstitial dumbbell mechanism in $M_3HCh$ as a function of the force constant $k_{rot}$ with respect to the $HM_6$ rotational motion. **d** Arrhenius plots of the total conductivity values for undoped $M_3HCh$ cold-pressed samples in the temperature range from 25 to 100 °C.

corresponds to the grain/bulk resistance and a low-frequency tail. As shown in Fig. 6b, the bulk activation energy for sodium-ion transport in $Na_{2.9}H(Se_{0.9}I_{0.1})$ is determined as $E_{a,bulk} = 0.18$ eV, which is close to the calculated migration barrier of 0.16 eV via vacancy mechanism in $Na_3HSe$ (Fig. 5a). Detailed structural characterization of $Na_{2.9}H(Se_{0.9}I_{0.1})$ is shown in Supplementary Fig. 20 and Supplementary Table 10.

The $^{23}Na$ NMR data (Fig. 6c) exhibit a distinct behavior associated with the bulk diffusion of $Na^+$ ions in pristine and I-doped samples. The analysis of the spectra revealed that quadrupole interaction is dominant in $Na_{2.9}H(Se_{0.9}I_{0.1})$ reflecting the faster $Na^+$ diffusion, whereas dipole–dipole interactions of the $^{23}Na$ nuclei are non-averaged in the isostructural $Na_3HSe$, $Na_3HTe$, and I-doped variant[52]. The line width of iodine-doped sample is narrower than that of undoped one, implying that faster short-range (i.e., bulk) ion dynamics. Furthermore, the line widths of $Na_3HTe$ NMR spectra (Supplementary Fig. 21) were observed to decrease with increasing temperature (known as

motional narrowing) and dipole–dipole interactions are progressively averaged due to the thermally activated motion of the Na ions[21]. The enhancement of $Na^+$ conductivity we have achieved by the common aliovalent substitution approach promises the excellent potential of soft hydride–chalcogenide anion framework for $Li^+/Na^+$ diffusion. There is still much room for improvement, such as mixed halide doping in A and/or B site, in future owing to the versatility of the perovskite-based crystal structures.

## Discussion

By exploiting the simple but unique hydride anion, a series of antiperovskites with the formula of $M_3HCh$ (M = Li, Na; Ch = S, Se, Te) has been successfully synthesized. Detailed structural characterizations demonstrate that the highly flexible size of hydride in the octahedral center explains the robustness of the ideal cubic structure in a wide compositional range with a 0.09 Å H–M bond difference, much longer than other antiperovskites

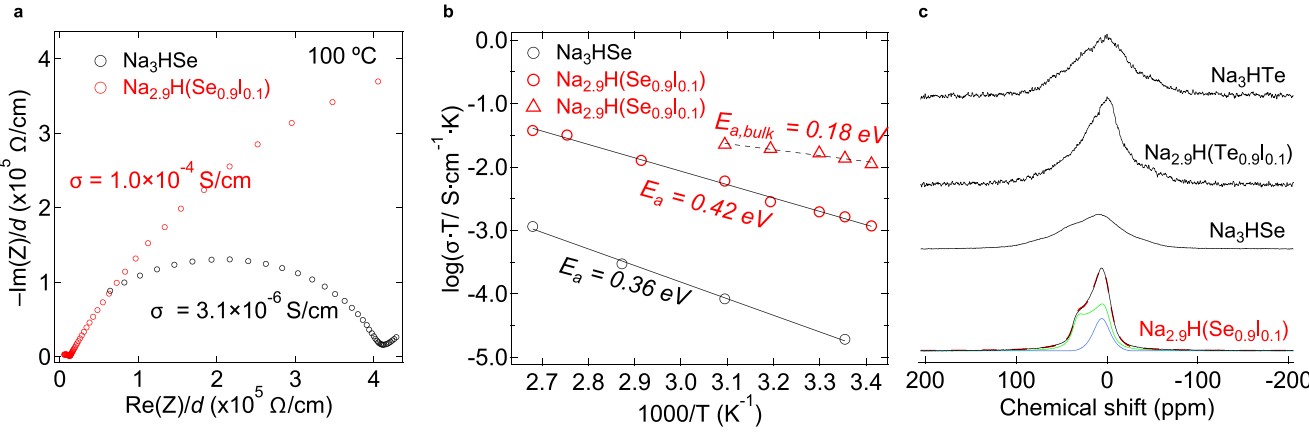

**Fig. 6 Na$^+$ diffusion in Na$_3$HSe and Na$_{2.9}$H(Se$_{0.9}$I$_{0.1}$). a** Impedance plots at 100 °C of the cold-pressed pellets and the impedance is normalized to the respective pellet thickness. **b** Arrhenius plots of the total conductivity (circles) and bulk conductivity (triangles). **c** $^{23}$Na static NMR spectra measured under 7 T at room temperature. For Na$_{2.9}$H(Se$_{0.9}$I$_{0.1}$), observed spectra (black solid), fitting spectra (red dashed), and deconvolved peak 1 with $C_Q = 1.30$ MHz, $\eta = 0.16$ (green) and peak 2 with $C_Q = 1.33$ MHz, $\eta = 0.03$ (blue).

and perovskites. Theoretically, the family of LiRAPs and NaRAPs composed by soft anion sublattices of polarizable hydride and chalcogenide are shown to have low migration barriers for Li/Na bulk transport, where the rotational phonon mode by HM$_6$ octahedron possibly facilitates the Li/Na migration. Experimentally, high conductivity and low bulk activation energy have been demonstrated in the iodine-doped Na$_{2.9}$H(Se$_{0.9}$I$_{0.1}$), with the total ionic conductivity of $1 \times 10^{-4}$ S/cm at 100 °C. Further optimization through structural design and chemical doping would promise to achieve their full potential. The exploitation of size-flexible hydride anion developed in this work will be favorable for other ion-conducting materials, also help to induce novel states of matter and excitation phenomena.

## Methods

**Materials synthesis**. Polycrystalline M$_3$HCh samples (M = Li, Na; Ch = S, Se, Te) were synthesized via high-pressure solid-state reactions using MH and M$_2$Ch as starting reagents. We used as-received LiH (Sigma-Aldrich, 99%), Li$_2$S (Sigma-Aldrich, 99%), NaH (Sigma-Aldrich, 99%), Na$_2$S (Sigma-Aldrich, 99%), Na$_2$Se (Kojundo, 99%), Na$_2$Te (Kojundo, 99%), while Li$_2$Se and Li$_2$Te were prepared using LiEt$_3$BH (Sigma-Aldrich), Se (Rare Metallic Co., LTD., 99.9%) and Te (Rare Metallic Co., LTD., 99.9%). There is a tendency that excess use of alkali metal hydride improves sample purity. The starting materials were well-ground and pelletized, and the pellets were inserted into a boron nitride (BN) sleeve. The two ends of the sleeve were then sealed with BN caps. The assemble was loaded into a graphite tube heater and then enclosed in a pyrophyllite cube serving as a pressure medium. All above procedures were handled in a N$_2$-filled glovebox due to air and moisture sensitivity of the precursors and synthesized materials. Finally, the assembly was pressed at 5 GPa, heated to 700 °C in 10 min, kept for 2 h, and cooled down to room temperature in 5 min before pressure was released. All the iodine-doped samples (e.g., Na$_{2.9}$H(Se$_{0.9}$I$_{0.1}$), Na$_{2.9}$H(Te$_{0.9}$I$_{0.1}$), Li$_{2.9}$H(S$_{0.9}$I$_{0.1}$)) were successfully synthesized using NaI/LiI (Sigma-Aldrich, 99%) as iodine source under the same condition with undoped one.

**Powder X-ray and neutron diffraction**. We characterized the purity and crystal structures of as-prepared M$_3$HCh samples at room temperature by powder X-ray diffraction (XRD) measurements using Rigaku Smart Lab with Cu radiation and Bruker D8 ADVANCE with Mo radiation. High-resolution synchrotron XRD experiments (SXRD) were performed at BL02B2 of SPring-8 (Japan) equipped with MYTHEN solid-state detectors. Time-of-flight(TOF) powder neutron diffraction (ND) data of Li$_3$HS and Na$_3$HS samples were collected on iMATERIA and SPICA diffractometers installed at the Material and Life science Facility (MLF) in the Japan Proton Accelerator Research Complex (J-PARC). The powder samples were sealed in cylindrical vanadium cells of dimensions 6 mm in diameter, 55 mm in height, and 100 μm in thickness. Rietveld refinements were performed on neutron data taken at the backscattering bank ($2\theta = 155°$ for iMATERIA and $2\theta = 160.77°$ for SPICA). Data were evaluated and refined using the FULLPROF suite, JANA2006, and Z-Rietveld softwares[53]. VESTA was used to display crystal structure and to calculate geometric properties.

**Impedance spectroscopy**. Ionic conductivities of cold-pressed pellets of M$_3$HCh were measured by electrochemical impedance spectroscopy (EIS) with a constant voltage of 10 mV in the frequency range of 1 MHz to 0.1 Hz using an ECS Modulab potentiostat/galvanostat. Around 80 mg of the specimen was placed between two stainless-steel rods as an ion-blocking electrode in a custom-made Swagelok cell and pressed into a 10 mm diameter pellet by a hydraulic press at 18 MPa for 1 min in an Ar-filled glovebox. For activation energy measurements, we applied a temperature loop starting from 25 °C to 100 °C. The measurements at each temperature were collected after being held for 3 h to ensure the temperature stabilization. All equivalent circuits of Nquist plots were fitted using the EC-Lab software package Z-fit.

**NMR spectroscopy**. $^{23}$Na NMR measurements were performed using JNM-ECA300 under 7 T with the $^{23}$Na Lamor frequency of 79.5 MHz and ECA600 (JEOL) under 14 T with 158.8 MHz frequency. For magic-angle spinning (MAS) NMR, the sample was packed into a 4-mm zirconia rotor in an Ar-filled glovebox and the spinning speed was 10 kHz. For variable-temperature $^{23}$Na static NMR, the sample was sealed in an evacuated borosilicate glass tube. The $^{23}$Na chemical shifts of all spectra were referenced to 1 M NaCl aqueous solution at 0 ppm and the peaks of the spectra were deconvoluted using a Dmfit program. To assign the observed $^{23}$Na spectra, first-principles calculations based on DFT were carried out using the WIEN2k codes[54,55]. The muffin-tin radius, $R_{MT}$, of Na, H, and Se atoms are 2.0, 1.5, and 2.0, respectively, for cubic Na$_3$HSe. The volume optimization was performed prior to the chemical shift calculation. Self-consistent cycles were carried out at an energy convergence of 0.0001 Ry. The $R_{MT}K_{max}$ determining the number of basis function was set to 7.0, $G_{max}$ was 12, and the number of $k$-points in the irreducible Brillouin zone was 35. The electric-field gradient (EFG), the asymmetry parameter ($\eta$), and the magnetic shielding coefficient ($\sigma_{iso}$) were calculated for the final optimized structure. To convert $\sigma_{iso}$ into the chemical shift ($\delta_{iso}$) for comparison, $\sigma_{iso}$ of Na$_2$SiO$_3$ with $\delta_{iso} = 20.0$ ppm[56], α-Na$_2$Si$_2$O$_5$ with $\delta_{iso} = 17.4$ ppm[52], and Na$_2$SO$_4$ with $\delta_{iso} = -8.5$ ppm[52] were calculated in the same manner.

## Data availability

The data that support the findings of this study are available from the corresponding author upon reasonable request.

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

## Acknowledgements

This work was supported by Grants-in-Aid for Scientific Research on Innovative Areas "Mixed Anion" (No. JP16H06439; JP16H06440; JP16H06441; JP17H05491), JSPS Core-to-Core Program (A) Advanced Research Networks (16H00888), CREST (JPMJCR1421), and JSPS KAKENHI (18H03832). The neutron experiments were conducted at J-PARC (2019A0017, 2017L1302). The synchrotron radiation experiments were performed at the BL02B2 of SPring-8, with the approval of the Japan Synchrotron Radiation Research Institute (JASRI).

## Author contributions

S.H.G., T.B., and S.F. contributed equally to this work. S.H.G. and T.B. carried out all the synthetic work. T.B., C.T., and H.U. contributed to the refinement of XRD and ND data. S.F. and A.K. performed theoretical calculations. S.H.G., K.Y., Y.X., and Y.U. contributed the conductivity measurements. I.O. and H.T. contributed the NMR measurements and analysis. Y.W. contributed to the precursor preparation. K.F. and M.Y. helped ND measurement and analyses. S.H.G., S.F., C.T., and H.K. prepared the paper and figures. H.K. designed and coordinated this study, contributed to all measurements and analyses.

## Competing interests

The authors declare no competing interests.
