## [Peer Review File · Nature Communications]

Reviewer #1 (Remarks to the Author):

The manuscript describes the structural and conducting properties of several M_3HCh antiperovskites, with $M = Li, Na$ and $Ch = S, Se, Te$. These are somewhat related to M_3OHal ($Hal = Cl, Br$) systems, which were originally reported to be impressive Li-ion conductors, but subsequently shown to possess significant H^+ conductivity arising from their moisture sensitivity. Nevertheless, the compounds reported in this manuscript are highly novel and, as the authors state towards the end of the text, will form parent compounds for further studies to optimise the conducting properties using aliovalent doping methods.

I recommend that the manuscript be published in Nature Comms, subject to the following revisions.

(i) The authors should explore the relationship between the perovskite and antiperovskite structures, including the tolerance factor, t . In the case of ABX_3 perovskites, the X ions are anions and, therefore, generally larger. Crudely, to achieve $t \sim 1$ requires the A and B cations to have significantly different sizes. However, for X_3BA antiperovskites, the smaller X cations means that the A and B anions tend to be more similar in size. An important consequence of this is that anti-site disorder of the A and B positions can occur. This has been discussed in the case of the Ag-ion conductor Ag_3SI , (see S. Hull et al, "The crystal structures of superionic Ag_3SI ", J. Phys.: Condens. Matter, 13, 2295-2316 (2001) and S. Hull et al, "Ionic diffusion within the a^* and b phases of Ag_3SI ", J. Phys.: Condens. Matter, 19, (2007)), where local disorder of the S²⁻ and I⁻ has profound influence on the Ag^+ diffusion. I recommend that the authors consider this possibility, either in their analysis of the diffraction data or theoretical calculations.

(ii) The authors should expand on the comment on page 11 that ambient temperature synthesis of Na_3HTe produced a sample with an "almost identical" X-ray diffraction pattern. Perhaps this is linked to the point above?

(iii) Given the confusion within the literature concerning the nature of the ionic conduction mechanism of the M_3OHal compounds, the authors should better justify (i.e. experimentally) why they attribute the high ionic conductivity solely to the motion of Li^+/Na^+ .

(iv) Perhaps the authors could comment on the highly anisotropic nature of the M^+ thermal vibrations shown in Supplementary Table 1, etc.? Is that related to the rotational phonon mode illustrated in Figure 4?

Reviewer #2 (Remarks to the Author):

In their paper, Gao et al. report a new class of antiperovskite ionic conductors based on hydride anions. Explorations of the materials chemistry reveal certain structures and compositions with very fast ion migration. Computational studies are used to explore the origins of the low migration barriers, which are attributed to a deformable lattice featuring polarizable hydride ions. This appears to be a promising class of solid electrolytes that merits further exploration, and from this point of view, the work certainly merits publication in some form. However, there are some issues that must be resolved first, including some apparent inconsistencies in the computed vs. measured results.

Overall, these inconsistencies make it difficult to determine the take-home conclusions should be, particularly for the computational contributions.

1) On p. 15, the authors state that “the low activation energies promising fast ionic transport in solid-state conductors conform to the calculated migration barriers for bulk ionic transport.” In fact, the difference between the computed and measured activation energies is somewhat troubling. The authors report experimental activation barriers from 0.44 to 0.53 eV (0.30 eV for Na₃HS), but the computed barriers are much lower (0.15 to 0.30 eV for vacancy mechanism and far lower for dumbbell). The dumbbell mechanism seems particularly unlikely here because its barriers are especially low. The authors attribute the discrepancy between experiment and theory to the presence of higher-barrier conduction channels along grain boundaries. Although this is possible, the measured barrier would imply that the grain boundaries are the dominant conduction pathways, in which case the crystalline models and mechanisms may have less relevance. Do the authors have any information regarding the microstructure of their materials that would suggest high density of grain boundary conduction pathways? If the dominance of grain boundaries can be explained by a low carrier density in the grains, then this should also be reflected in the prefactors obtained from the Arrhenius plots of the ionic conductivity.

2) Similarly, even if the absolute ionic conductivities are not correctly predicted, one might assume that the trends should follow the predictions (for instance, local lattice softening and facile rotation would likely still play some role even at grain boundaries). However, this does not seem to be the case. For instance, Li₃HTe has the highest computed barrier but among the lowest barriers and highest ionic conductivities experimentally. I suspect that something fundamental is missing in the interpretation of the conduction mechanism. By analogy with Li₂HOCl, defects may change the mechanism considerably. I realize that a full exploration is beyond the scope of the current paper, but the authors should devote some effort to identifying possibilities, as this could be critical for proper interpretation of mechanisms and guidance for future development.

3) In determining the “size change” of the hydride in the different compounds, the calculation relies on fixing the ionic radii of the other species. However, other atoms’ sizes can also change upon incorporation of other atomic species. How can these be taken into account? Similarly, the hydride is not the only polarizable species that could be playing a role in accommodating rotation or diffusion-- in particular, the sulfide is more polarizable. Is there any understanding (perhaps from the calculations) of the role of the sulfide?

4) The authors attribute the conduction mechanism to enhanced rotational mobility of hydride complexes due to hydride polarizability/deformability. A great deal of discussion is devoted to this point in the main text, but the most convincing results along these lines are found in Figure S14. I recommend moving this figure into the main text, along with an expanded discussion of these data.

5) Can the authors verify that the system is a single-ion Li⁺/Na⁺ conductor? In particular, can they show that hydride ions are not migrating?

6) It is sometimes difficult to understand which results are computed and which are measured (for example, in Fig. 5). The authors should be careful to clearly distinguish the two. Similarly, it is sometimes confusing which computational results correspond to the cubic Na₃HS versus the orthorhombic structure.

7) On p.2-3, the authors cite two features of the anion-host matrix that can contribute to fast ionic conductivity. In practice, there are many such features that have been proposed. One of the most relevant here would seem to be the soft rotational mobility, which isn't specifically highlighted in the introduction but has been discussed at length for other solid-state conductors (DOI: 10.1016/j.chempr.2019.07.001; DOI: 10.1021/acs.chemmater.7b02902; DOI: 10.1002/anie.199115471; DOI: 10.1021/acs.chemmater.9b01435; DOI: 10.1038/s41467-020-15245-5). It may also be useful to cite some of the recent reviews on this topic, which have much more comprehensive discussions (DOI: 10.1021/acs.chemrev.5b00563; DOI: 10.1021/acs.chemrev.9b00747; DOI: 10.1088/2516-1083/ab73dd).

8) Some acronyms (e.g., ND) aren't defined until the Methods section, which appears at the end.

Reviewer #3 (Remarks to the Author):

This paper considers the incorporation of hydride anions in mixed anion anti-perovskites and reports the high pressure synthesis of a new family of alkali ion conducting phases. The concept is interesting and the syntheses successful but unfortunately, the materials are not good ionic conductors. With usual units of S/cm (rather than mS/cm K which are used in the paper) conductivity values at room temperature are in the range 10^{-8} to 10^{-10} , which are many orders of magnitude lower than the solid electrolytes considered for battery applications, eg 10^{-1} to 10^{-3} for beta aluminas, 10^{-2} to 10^{-3} for sulphide thiolisicons and 10^{-3} to 10^{-5} for Li garnets.

The authors use the terms 'fast ion conductors' and 'superionic conductors', which are now regarded as misnomers, to describe their materials, but anyway, they will be of minimum interest to the electrolyte community unless the conductivities can be increased by several orders of magnitude with appropriate doping.

The structural aspects of the paper are interesting but much of the content on electrical conductivity is not really relevant given the poor conductivity of the materials.

Reviewer #1

The manuscript describes the structural and conducting properties of several M₃HCh antiperovskites, with M= Li, Na and Ch = S, Se, Te. These are somewhat related to M₃OHal (Hal=Cl, Br) systems, which were originally reported to be impressive Li-ion conductors, but subsequently shown to possess significant H⁺ conductivity arising from their moisture sensitivity. Nevertheless, the compounds reported in this manuscript are highly novel and, as the authors state towards the end of the text, will form parent compounds for further studies to optimise the conducting properties using aliovalent doping methods.

I recommend that the manuscript be published in Nature Comms, subject to the following revisions.

Response:

We thank the referee for the supportive assessments of this new series of hydride-based anti-perovskites and noticing a critical issue on currently reported anti-perovskite ionic conductors (M₃OX (X = Cl, Br) systems).

(i) The authors should explore the relationship between the perovskite and antiperovskite structures, including the tolerance factor, t . In the case of ABX₃ perovskites, the X ions are anions and, therefore, generally larger. Crudely, to achieve $t \sim 1$ requires the A and B cations to have significantly different sizes. However, for X₃BA antiperovskites, the smaller X cations means that the A and B anions tend to be more similar in size. An important consequence of this is that anti-site disorder of the A and B positions can occur. This has been discussed in the case of the Ag-ion conductor Ag₃SI, (see S. Hull et al, "The crystal structures of superionic Ag₃SI", J. Phys.: Condens. Matter, 13, 2295-2316 (2001) and S. Hull et al, "Ionic diffusion within the a and b phases of Ag₃SI", J. Phys.: Condens. Matter, 19, (2007)), where local disorder of the S²⁻ and I⁻ has profound influence on the Ag⁺ diffusion. I recommend that the authors consider this possibility, either in their analysis of the diffraction data or theoretical calculations.*

Response:

We thank the referee for pointing out the possibility of antisite disorder of anions in our compounds, in view of antisite disorder cases of Ag₃SI which we were unaware of. We have performed Rietveld refinement on the time-of-flight neutron powder diffraction data of Li₃HS, assuming the antisite disorder in the 'stoichiometric' composition (where $g_A = g_A(S) + g_A(H)$, $g_B = g_B(S) + g_B(H)$, $g_A(H) + g_B(H) = 1$, and $g_A(S) + g_B(S) = 1$), and obtained $g_B(H) = 1.015(5)$ and $g_A(H) = -0.015(5)$ with $R_{wp} = 1.37\%$ and $R_{Bragg} = 6.64\%$. This result suggests that the antisite disorder, if it exists, is negligibly minimal. Note that our original refinement without antisite disorder (Supplementary Table 1) gave similar reliability factors of $R_{wp} = 1.35\%$ and $R_{Bragg} = 6.28\%$. Likewise, the refinements of synchrotron X-ray diffraction (SXRD) data of Na₃HS, and Na₃HSe and Na₃HTe reveal no appreciable antisite disorder. We have included the discussion on anti-site disorder accordingly in *Structure Determination*

section and details in the corresponding Supplementary Tables 1-4 which list the refinement results.

Additionally, the sharp peak (centered at 24.2 ppm) in ^{23}Na magic-angle-spinning (MAS) NMR spectra of Na_3HSe , conducted by newly added co-authors, Drs. I. Oikawa and H. Takamura, also implies sodium atom is in a very similar environment, in agreement with the refinement result that no significant chemical disorder occurs in Na_3HSe (Supplementary Figure 5 and Supplementary Table 2). The new NMR information has been provided in the revised manuscript with some additional figures (Figure 6c and Supplementary Figure 7).

(ii) The authors should expand on the comment on page 11 that ambient temperature synthesis of Na_3HTe produced a sample with an “almost identical” X-ray diffraction pattern. Perhaps this is linked to the point above?

Response:

We admit the vague description of X-ray diffraction data of Na_3HTe obtained under ambient pressure, since only a Le Bail analysis was performed at that time for its laboratory X-ray diffraction profile, resulting in the same lattice parameters as the sample prepared under pressure within the errors. The absence of *Rietveld* analysis led us to describe “almost identical” in the original manuscript. We have followed the suggestion by the reviewer and newly collected the data using synchrotron X-ray source. A *Rietveld* refinement of the ambient phase of Na_3HTe shows that the structure is indeed identical with the one prepared under high-pressure (5 GPa). We have expanded the explanation regarding the structure analysis of ambient phase and added the *Rietveld* refinement results (e.g., displacement parameters and occupancy factors) in the Supplementary Figure 12 and Supplementary Table 8.

(iii) Given the confusion within the literature concerning the nature of the ionic conduction mechanism of the $M_3\text{OHal}$ compounds, the authors should better justify (i.e. experimentally) why they attribute the high ionic conductivity solely to the motion of Li^+/Na^+ .

Response:

Thank you for bringing in this point to us. The recent paper (Ref. 17; *Chem. Mater.* 2018, 30, 8134) has pointed out the inclusion of protons (or hydroxide groups), questioned the existence of proton-free “ Li_3OCl ”, and called the ionic nature of lithium oxyhalides into questions. Thus, the referee might suspect that our $M_3\text{HCh}$ compounds contain protons. However, this is unlikely because the samples of “ Li_3OCl ” are reportedly synthesized using LiCl and LiOH as starting materials, whereas in our case of $M_3\text{HCh}$ we use lithium/sodium hydrides and chalcogenides to strictly avoid the influence of moisture or humidity. Furthermore, under reactive atmosphere during the synthesis process, there is hardly any chance of hydrides being oxidized (to H^+), which precludes the presence of the proton and the possibility of proton conductivity.

Perhaps the referee is also wondering about the possibility of hydride motion; $M_3\text{HCh}$ contains more than one dynamic species (H^- and Li^+/Na^+). Reviewer 2 also pointed out this issue. We have taken this into account by calculating the migration

energy barriers of H^- in comparison with that of Li^+ in Li_3HS framework. In Supplementary Figure 16 (in revised Supplementary Information), the barrier for H^- motion via vacancy mechanism is calculated to be 3.49 eV, which is much higher than 0.20 eV for Li^+ migration energy, indicating that the diffusion of H^- ions is energetically unfavorable. Additionally, no direct/straight migration path between neighboring H^- sites (H-H distance of ~ 3.85 Å in Li_3HS) exists and the bottleneck between Li^+ and S^{2-} is too small for large H^- to migrate. To further confirm the charge carriers, we have conducted a potentiostatic measurement of a symmetric $\text{Li}_3\text{PS}_4/\text{Li}_{2.9}\text{H}(\text{S}_{0.9}\text{I}_{0.1})/\text{Li}_3\text{PS}_4$ cell at room temperature, with as-synthesized iodine-doped Li_3HS as the solid electrolyte, Li-ion conductor Li_3PS_4 as the working electrode and the counter electrode. When applied a DC voltage of 0.5 V, as shown in Supplementary Figure 17, a steady current is observed, which suggests the Li^+ is migrating in the iodine doped Li_3HS .

We have included these arguments in the main text's *Ionic conductivity* section following the discussion on the calculated migration barrier for cationic transport. Two figures (Supplementary Figure 16 showing calculated migration barrier of each ion in Li_3HS and Supplementary Figure 17 displaying the current curve of $\text{Li}_{2.9}\text{H}(\text{S}_{0.9}\text{I}_{0.1})$) have been added.

(iv) Perhaps the authors could comment on the highly anisotropic nature of the M -thermal vibrations shown in Supplementary Table 1, etc.? Is that related to the rotational phonon mode illustrated in Figure 4?

Response:

Thank you for bringing this point to us. The large anisotropic displacement parameters of Li^+/Na^+ for our $M_3\text{HCh}$ compounds is not uncommon to other solid-state electrolytes, for instance, the superionic conductor $\text{Li}_{10}\text{GeP}_2\text{S}_{12}$ (Ref. 32, *Nat. Mater.* 2011, 10, 682–686).

As the referee points out, the anisotropic broadening parameters of the alkali atoms are to a large extent correlated to the rotational phonon mode reflecting at the M point in the Brillouin zone (Figure 4d in the revised manuscript). The M -point phonon mode related with the octahedral tilting in high-symmetry cubic oxide perovskites is extremely soft and dynamic in our hydride antiperovskite. The Rietveld refinement of neutron data of Li_3HS showed that the displacement ellipsoids of the alkali elements in the cubic phases, U_{22} and U_{33} , are much larger than U_{11} , as displayed below. The observed thermal ellipsoids (see figure below) conform well to octahedral rotation associated with the M point.

We have included this figure in Figure 4e and briefly extended in the discussion on the rotational phonon mode.

Reviewer #2

In their paper, Gao et al. report a new class of antiperovskite ionic conductors based on hydride anions. Explorations of the materials chemistry reveal certain structures and compositions with very fast ion migration. Computational studies are used to explore the origins of the low migration barriers, which are attributed to a deformable lattice featuring polarizable hydride ions. This appears to be a promising class of solid electrolytes that merits further exploration, and from this point of view, the work certainly merits publication in some form. However, there are some issues that must be resolved first, including some apparent inconsistencies in the computed vs. measured results. Overall, these inconsistencies make it difficult to determine the take-home conclusions should be, particularly for the computational contributions.

Response:

We appreciate Reviewer 2's comments and recognition of the importance of our work. We hope the current changes address the reviewer's concerns.

1) On p. 15, the authors state that "the low activation energies promising fast ionic transport in solid-state conductors conform to the calculated migration barriers for bulk ionic transport." In fact, the difference between the computed and measured activation energies is somewhat troubling. The authors report experimental activation barriers from 0.44 to 0.53 eV (0.30 eV for Na₃HS), but the computed barriers are much lower (0.15 to 0.30 eV for vacancy mechanism and far lower for dumbbell). The dumbbell mechanism seems particularly unlikely here because its barriers are especially low. The authors attribute the discrepancy between experiment and theory to the presence of higher-barrier conduction channels along grain boundaries. Although this is possible, the measured barrier would imply that the grain boundaries are the dominant conduction pathways, in which case the crystalline models and mechanisms may have less relevance. Do the authors have any information regarding the microstructure of their materials that would suggest high density of grain boundary conduction pathways? If the dominance of grain boundaries can be explained by a low carrier density in the grains, then this should also be reflected in the prefactors obtained from the Arrhenius plots of the ionic conductivity.

Response:

As the referee points out, the relatively large activation energies extracted from total ionic conductivity of these pristine (i.e., stoichiometric) compounds differ from their computed migration barriers for a vacancy/interstitial mechanism.

Regarding the microstructure, all the materials are prepared in similar conditions using high-pressure and high-temperature synthesis, therefore we expect that their microstructures should be similar and not have any unusually high surface areas (e.g.,

nanostructures). We considered measuring scanning electron microscopy on the samples, however, they are readily oxidized in air.

The discrepancy can rather be explained quite adequately by two reasons: the low concentrations of intrinsic charge carriers in the pristine compounds, however the calculated migration barrier assumes intrinsic vacancies/interstitials of lithium/sodium. Additionally, the experimentally obtained activation energy takes the defect formation energy into account as well, whereas the calculated migration barrier is separate from the defect formation energy. Actually, the defect formation energies (e.g., in Li_3HS , the formation energies of the LiH Schottky pair and the Li_2S Schottky pair are 0.707 eV and 0.653 eV, respectively) calculated by DFT are relatively high.

To probe the short-range or bulk ion dynamics (e.g., diffusion coefficient, activation energy), we have conducted nuclear magnetic resonance (NMR) spin-lattice relaxation (SLR) T_1 , but could not observe a minimum versus $1/T_1$ in the recorded temperature range of 25 to 320 °C and then the activation energy of the mobile ions.

Alongside the improvement of their ionic conductivities by chemical doping, the bulk ionic conductivity is now accessible in a lower temperature range. The Nyquist plots of iodine-doped $\text{Na}_{2.9}\text{H}(\text{Se}_{0.9}\text{I}_{0.1})$ at lower temperatures show two semicircles in which a small, poorly resolved semicircle in the high-frequency region corresponding to bulk transport (as shown in Supplementary Figure 19). The activation energy from the Arrhenius plot (Figure 6b) of the bulk ionic conductivity of $\text{Na}_{2.9}\text{H}(\text{Se}_{0.9}\text{I}_{0.1})$ is determined as 0.185 eV, which is comparable to the calculated migration barrier of 0.16 eV for bulk Na^+ diffusion via a vacancy mechanism. This consistency suggests that the transport mechanisms we have proposed are valid and the calculation results are helpful for further enhancement of the ionic conductivity. We have also attempted iodine substitution for chalcogenide in $\text{Li}_{2.9}\text{H}(\text{S}_{0.9}\text{I}_{0.1})$, $\text{Li}_{2.9}\text{H}(\text{Se}_{0.9}\text{I}_{0.1})$, $\text{Li}_{2.9}\text{H}(\text{Te}_{0.9}\text{I}_{0.1})$ and $\text{Na}_{2.9}\text{H}(\text{Te}_{0.9}\text{I}_{0.1})$ and all of them show much higher conductivities than the undoped ones. Though, the grain contribution cannot be deconvoluted/detected from the total resistance within the frequency scale of the current instrumentation. We believe that, as with $\text{Na}_{2.9}\text{H}(\text{Se}_{0.9}\text{I}_{0.1})$, there is still much room for further improvements and other type of aliovalent ion substitution (Cl^- for S^{2-} and/or Ca^{2+} for Na^+) in future would not only greatly enhance the conductivities but probably also confirm the calculated low bulk migration barrier.

The discussion on EIS and NMR data has been significantly updated in the *Ionic Conductivity* section and new Figure 6 has been added in the main text accordingly. We have added new co-authors (Prof. Hitoshi Takamura and Dr. Itaru Oikawa from Tohoku University) who conducted NMR experiments. All other co-authors agree with this addition.

2) Similarly, even if the absolute ionic conductivities are not correctly predicted, one might assume that the trends should follow the predictions (for instance, local lattice softening and facile rotation would likely still play some role even at grain boundaries). However, this does not seem to be the case. For instance, Li_3HTe has the highest computed barrier but among the lowest barriers and highest ionic conductivities experimentally. I suspect that something fundamental is missing in the interpretation

of the conduction mechanism. By analogy with Li₂HOCI, defects may change the mechanism considerably. I realize that a full exploration is beyond the scope of the current paper, but the authors should devote some effort to identifying possibilities, as this could be critical for proper interpretation of mechanisms and guidance for future development.

Response:

We are grateful for the referee's considerate and pertinent comments. Perhaps the referee is concerned about the discrepancy between theoretical (Figure 5a and 5b) and experimental (Figure 5c) results for activation energy values and trends. As we have explained in the first comment, the calculated migration barrier is based on the vacancy mechanism or interstitial dumbbell mechanism, but it doesn't take the defect (vacancy/interstitial) formation energy into account. In this respect, we could expect that when the Li/Na vacancies are introduced by substituting chalcogenide anions with monovalent iodine, the experimental activation energy becomes close to the migration barrier.

In the case of Na_{2.9}H(Se_{0.9}I_{0.1}), the extracted grain/bulk activation energy of 0.185 eV is in good agreement with the calculated migration barrier of 0.16 eV (for Na₃HSe via vacancy mechanism), which validates our proposed migration model and mechanism. Using the same strategy to other compounds in this series of antiperovskite, the high ionic conductivity has been achieved in every composition, yet we could not resolve between the grain/bulk and grain boundary semicircles and then clarify the trends in the series of antiperovskites) within the frequency scale of the current instrumentation. For the pristine compounds (e.g., Li₃HTe), we still have little knowledge of the deviation from the predicted trends or some other factors underlying the diffusion process. We appreciate the referee's consideration that a full exploration on the conduction mechanism (e.g., the impact of local softness on the grain boundary resistance) is beyond the scope of the current paper.

This has been addressed in the discussion of ionic conductivity properties with additional supplementary figures and Figure 6 in main text. Hopefully the manuscript has become clearer as a result.

3) In determining the "size change" of the hydride in the different compounds, the calculation relies on fixing the ionic radii of the other species. However, other atoms' sizes can also change upon incorporation of other atomic species. How can these be taken into account? Similarly, the hydride is not the only polarizable species that could be playing a role in accommodating rotation or diffusion--in particular, the sulfide is more polarizable. Is there any understanding (perhaps from the calculations) of the role of the sulfide?

Response:

We thank the reviewer for taking a critical view of the 'polarizable' hydride anion, noting that in our compounds the chalcogen is also a polarizable species. As the referee notes, when determining the "size change" of the hydride in the series of lithium- or sodium-rich antiperovskites, we fixed the ionic radii of Li⁺/Na⁺ and calculated the "real" ionic radius of hydride. This analysis stems from our recent work on

oxyhydrides (e.g., *Nat. Commun.* 2017, 8, 1217), and has been utilized to prove the size flexibility of H^- in our antiperovskites. However, we admit that this method does not allow simultaneous estimation of the size change of chalcogenide anions.

We have followed the suggestion by the reviewer to conduct a grid-based Bader analysis (Ref. 44) to theoretically examine the size change of each ion in our antiperovskites. The resulting “Bader radius” of Li^+ ion, which is extracted from Bader volume of charge density grid of ion, only increases from 0.94 to 0.97 Å when the A-site chalcogen ion increasing from S^{2-} to Te^{2-} , whereas the Bader radius of H^- ion expands from 1.38 to 1.48 Å, as expected. The same as well holds true for the Na-rich hydride-chalcogenides. We note that the Bader radii are larger than the widely used Shannon ionic radii which we used for the size determination in the main text. It turns out that the change in the Bader radius of Ch^{2-} ions is comparable to that of H^- ions; The Bader radii of S^{2-} and H^- is 1.97 and 1.38 Å, respectively, in Li_3HS , and increased significantly to 2.15 and 1.48 Å in Na_3HS . In this respect, the robustness of cubic symmetry for $M_3\text{HCh}$ compounds could be attributed to the simultaneous and substantial expansion of HM_6 and ChM_{12} polyhedra (as shown in Supplementary Figure 10). This actually provides one answer to the reviewer’s question that “*Is there understanding of the role of the sulfide?*” As already illustrated before in the Supplementary Figure 14 (now we put it into Figure 5c in the main text following the suggestion of comment 4), the chalcogen ions do play a role in Li^+/Na^+ migration via rotational phonon mode. The rotational phonon mode at the M point (see Figure 4b and 4c) is indicative of the dynamic stability of cubic symmetry and also may be pertinent to ionic diffusion as Figure 5c shows that the Li^+/Na^+ migration barriers increase in the order of S, Se and Te.

Following the reviewer’s enlightening comment, we further investigated the Bader radius of H^- and Ch^{2-} ions under external pressure, and found that H^- ions are more sensitive (flexible) to the applied pressure than Ch^{2-} ions; When pressure is applied to 5 GPa, the size of H^- ion decreases more substantially than that of Ch^{2-} in all examined $M_3\text{HCh}$ antiperovskites. Thus, our initial claim that flexible hydride ions are critical for stabilizing the cubic symmetry of $M_3\text{HCh}$ antiperovskite is cogent, as most of $M_3\text{HCh}$ compounds prepared here are synthesized by high-pressure route. The peculiar pressure dependence of hydride size is also observed in our previous study on LaHO (Ref. 41; *J. Am. Chem. Soc.* 2019, 141, 871).

We have added a new paragraph regarding the Bader radii of chalcogenide following behind the discussion on the Shannon ionic size change of hydride and the detailed calculations of Bader radius into *Supplementary Information* with Supplementary Figure 11.

4) The authors attribute the conduction mechanism to enhanced rotational mobility of hydride complexes due to hydride polarizability/deformability. A great deal of discussion is devoted to this point in the main text, but the most convincing results along these lines are found in Figure S14. I recommend moving this figure into the main text, along with an expanded discussion of these data.

Response:

We appreciate the referee's nice suggestion. We have aligned this figure into the *main text* (Figure 5c) and updated the discussion accordingly.

5) *Can the authors verify that the system is a single-ion Li⁺/Na⁺ conductor? In particular, can they show that hydride ions are not migrating?*

Response:

Following the suggestion by both Reviewer 1 and Reviewer 2, we have carried out DFT calculation and electrochemical experiment to show that hydride is not migrating in our compounds. Theoretically, we calculated the migration energy barriers of H⁻ in the Li₃HS framework and compared with that of Li⁺. The barrier for H⁻ motion via vacancy mechanism is calculated to be 3.49 eV, which is far higher than 0.20 eV of Li⁺ migration energy, indicating that H⁻ ion diffusion is quite unlikely. Additionally, there is no direct/straight migration path between neighboring H⁻ sites (H⁻ sits in the center of the octahedron), and the bottleneck between Li⁺ and S²⁻ is too small to migrate for large H⁻ ions.

Experimentally, we have conducted a potentiostatic measurement of a symmetric Li₃PS₄/Li_{2.9}H(S_{0.9}I_{0.1})/Li₃PS₄ cell at room temperature, with as-synthesized iodine-doped Li₃HS as the solid electrolyte, Li-ion conductor Li₃PS₄ as the working electrode and the counter electrode. When applied a DC voltage of 0.5 V, as shown in Supplementary Figure 17, a steady current is observed, which suggests the Li⁺ is migrating in the iodine doped Li₃HS.

We have added these computational and experimental results to the *Supplementary Information*, and included a discussion in the main text's *Ionic conductivity* section.

6) *It is sometimes difficult to understand which results are computed and which are measured (for example, in Fig. 5). The authors should be careful to clearly distinguish the two. Similarly, it is sometimes confusing which computational results correspond to the cubic Na₃HS versus the orthorhombic structure.*

Response:

We apologize for the confusion. The manuscript has been modified to clearly distinguish the experimental and theoretical data.

7) *On p.2-3, the authors cite two features of the anion-host matrix that can contribute to fast ionic conductivity. In practice, there are many such features that have been proposed. One of the most relevant here would seem to be the soft rotational mobility, which isn't specifically highlighted in the introduction but has been discussed at length for other solid-state conductors (DOI:10.1016/j.chempr.2019.07.001; DOI: 10.1021/acs.chemmater.7b02902; DOI:10.1002/anie.199115471; DOI: 10.1021/acs.chemmater.9b01435; DOI: 10.1038/s41467-020-15245-5). It may also be useful to cite some of the recent reviews on this topic, which have much more comprehensive discussions (DOI: 10.1021/acs.chemrev.5b00563; DOI: 10.1021/acs.chemrev.9b00747; DOI: 10.1088/2516-1083/ab73dd).*

Response:

We appreciate the helpful comments. We have included all these references highlighting the soft rotational mobility in *Introduction* section (Refs. 18–22, 29–30).

8) *Some acronyms (e.g., ND) aren't defined until the Methods section, which appears at the end.*

Response:

We have corrected this mistake and defined all the acronyms when they first appear.

Reviewer #3

This paper considers the incorporation of hydride anions in mixed anion anti-perovskites and reports the high pressure synthesis of a new family of alkali ion conducting phases. The concept is interesting and the syntheses successful but unfortunately, the materials are not good ionic conductors. With usual units of S/cm (rather than mS/cm K which are used in the paper) conductivity values at room temperature are in the range 10^{-8} to 10^{-10} , which are many orders of magnitude lower than the solid electrolytes considered for battery applications, eg 10^{-1} to 10^{-3} for beta aluminas, 10^{-2} to 10^{-3} for sulphide thiolisicons and 10^{-3} to 10^{-5} for Li garnets.

The authors use the terms 'fast ion conductors' and 'superionic conductors', which are now regarded as misnomers, to describe their materials, but anyway, they will be of minimum interest to the electrolyte community unless the conductivities can be increased by several orders of magnitude with appropriate doping.

The structural aspects of the paper are interesting but much of the content on electrical conductivity is not really relevant given the poor conductivity of the materials.

Response:

We appreciate the reviewer for interests in structural characteristics of this new series of antiperovskites and recognition of conceptual novelty of our work. We acknowledge that the unsatisfying ionic conductivity of our pristine $M_3\text{HCh}$ compounds lags behind the well-known solid-state lithium-ion conductors (e.g., thio-LISICON $\text{Li}_{10}\text{GeP}_2\text{S}_{12}$, argyrodite $\text{Li}_6\text{PS}_5\text{Br}$, and garnet $\text{Li}_{6.55}\text{La}_3\text{Zr}_2\text{Ga}_{0.15}\text{O}_{12}$). We have followed the request from the reviewer to attempt reaching higher ionic conductivity through A-site doping with the iodine ion and successfully achieved significant enhancement of the total ionic conductivity in every iodine doped hydride-based antiperovskite (see figure below). We mainly prepared iodine-substituted samples (e.g., $\text{Li}_{2.9}\text{H}(\text{S}_{0.9}\text{I}_{0.1})$, $\text{Li}_{2.9}\text{H}(\text{Se}_{0.9}\text{I}_{0.1})$, $\text{Li}_{2.9}\text{H}(\text{Te}_{0.9}\text{I}_{0.1})$, $\text{Na}_{2.9}\text{H}(\text{S}_{0.9}\text{I}_{0.1})$, $\text{Na}_{3-x}\text{H}(\text{Se}_{1-x}\text{I}_x)$ with $x = 0.08, 0.10, 0.15, 0.20$, $\text{Na}_{2.9}\text{H}(\text{Te}_{0.9}\text{I}_{0.1})$), and all the derivatives provide about 2-3 orders of improvement in ionic conductivity over their parent stoichiometric compounds. In particular, $\text{Li}_{2.9}\text{H}(\text{S}_{0.9}\text{I}_{0.1})$ exhibits the high lithium-ion conductivity of 1.4×10^{-4} S/cm at 100 °C. $\text{Na}_{2.9}\text{H}(\text{Se}_{0.9}\text{I}_{0.1})$ also shows the high sodium-ion conductivity of 1.0×10^{-4} S/cm at 100 °C.

Arrhenius plots of the total conductivity for iodine doped (left) and undoped M_3HCh (right) cold-pressed samples in the temperature range from 20 to 100 °C.

We would like to point out that there is still much room to be explored in future owing to the flexibility of their intrinsic perovskite-type structure. It is feasible to control the local structure features by introducing mixed valence in both/either the A and/or B site for further enhancing favorable lithium/sodium ionic diffusion pathway. Thus, we are confident in this new series of hydride-based antiperovskites that not only have already revealed great potential to be developed into high-conductivity solid electrolytes, but also would offer various routes to investigate the underlying structure-property relationships in ionic conductors.

The fruitful optimization of conductivity we have achieved in the last two months attests to our concept and implies that the capacity of our soft/polarizable hydride-chalcogenide framework for Li^+/Na^+ diffusion. Finally, we would like to emphasize that alongside the high ionic conductivity, the structural features, such as the robustness of the ideal cubic structure and the presence of a soft phonon mode associated with HCh_6 octahedral rotation, are quite unique from the viewpoint of solid state chemistry and provide important insight towards the understanding of ionic transport. The approach to a flattened energy landscape by incorporating the hydride anion can be extended to other ion-conducting systems, such as the structurally similar perovskite oxygen-ion conductors of interest in solid-oxide fuel cells as well as other superionic conductors. We believe this context is what makes the work significant for *Nature Communications* readers with diverse disciplines.

Reviewer #1 (Remarks to the Author):

I have read the authors' responses to my original review and the modified version of the manuscript. I am happy that they have addressed the points made in my report and I recommend that the paper now be published in Nature Communications.

Reviewer #2 (Remarks to the Author):

The authors have done a good job of thoughtfully addressing my critiques and those of the other reviewers. The addition of the NMR results in particular are to be commended, as they appear to validate the interpretation of the discrepancy between the experimental and computational data in terms of the migration versus formation energy barriers. Notably, the fact that the formation energy is much higher than the migration barrier strengthens the notion that further improvement might be possible through ion substitution, as the authors suggest. Also, the additional analysis of the atomic radii based on Bader decomposition is a reasonable and physically justifiable method for reporting the size change and polarizability of the ions. I recommend that the manuscript be published without further revision.